

# Sources of PM$_{2.5}$ carbonaceous aerosol in Riyadh, Saudi Arabia

*Qijing Bian[1], Badr Alharbi[2], Mohammed M. Sharee[3], Tahir Husai[3], Mohammad J. Pasha[1], Samuel A. Atwood[1], Sonia M. Kreidenweis[1,*]*

[1]Department of Atmospheric Science, Colorado State University, Fort Collins, CO, 80526, USA.

[2]National Center for Environmental Technology, King Abdulaziz City for Science and Technology, P.O. Box 6086, Riyadh 11442, Saudi Arabia.

[3]Faculty of Engineering and Applied Science, Memorial University, St. John's, NL, A1B 3X5 Canada.

*Corresponding author: bianqj@atmos.colostate.edu and sonia@atmos.colostate.edu

## Abstract

Knowledge of the sources of carbonaceous aerosol affecting air quality in Riyadh, Saudi Arabia is limited, but needed for the development of pollution control strategies. We conducted sampling of PM$_{2.5}$ from April to September, 2012 at various sites in the city, and used a thermo-optical semi-continuous method to quantify the organic carbon (OC) and elemental carbon (EC) concentrations. The average OC and EC concentrations were 4.7 ± 4.4 and 2.1 ± 2.5 µg m$^{-3}$, respectively, during this period. Both OC and EC concentrations had strong diurnal variations, with peaks at 6-8 am and 20-22 pm, attributed to the combined effect of increased vehicle emissions during rush hour and the shallow boundary layer in the early morning and at night. This finding suggests a significant influence of local vehicular emissions on OC and EC. The OC/EC ratio in primary emissions was estimated to be 1.01, close to documented values for diesel emissions. Estimated primary (POC) and secondary (SOC) organic carbon concentrations were comparable, with average concentrations of 2.0 ± 2.4 and 2.8 ± 3.4 µg m$^{-3}$, respectively.

We also collected 24 hour samples of PM$_{10}$ onto quartz microfiber filters and analyzed these for an array of metals by ICP-OES. Total OC was correlated with Ca (R$^2$ of 0.63), suggesting they may have similar sources. In addition to a ubiquitous dust source, Ca is emitted during desalting processes in the numerous refineries in the region and from cement kilns, suggesting these sources may also contribute to observed OC concentrations in Riyadh. Concentration weighted trajectory (CWT) analysis showed that





high OC and EC concentrations were associated with air masses arriving from the Persian Gulf and the region around Baghdad, locations with high densities of oil fields and refineries as well as a large Saudi Arabian cement plant. We further applied positive matrix factorization to the aligned data set of EC, OC and metal concentrations (Al, Ca, Cu, Fe, K, Mg, Mn, Na, Ni, Pb and V). Three factors were derived, and were proposed to be associated with oil combustion, industrial emissions (Pb-based), and a combined source from oil fields, cement production, and local vehicular emissions. The dominant OC and EC source was the combined source, contributing 3.9 µg m$^{-3}$ (80%) to observed OC and 1.9 µg m$^{-3}$ (92%) to observed EC.

## 1. Introduction

Organic carbon (OC) and elemental carbon (EC) (or black carbon, BC, operationally identified based on detection method) are key components of the atmospheric aerosol (Jacobson et al., 2000). The contribution of carbonaceous components to total particulate matter (PM) concentrations varies with site and season, comprising from 20 to 90% of the total mass (Kanakidou, et al., 2005). EC is emitted from a variety of combustion processes (Bond et al., 2013), is classified as a short-lived climate forcer that contributes to atmospheric warming (Ramanathan and Carmichael, 2008), and is also associated with human morbidity and mortality (Weinhold, 2012). OC includes both direct emissions (primary organic carbon, POC) and secondary OC (SOC) formed in the atmosphere via oxidation (Robinson et al., 2007). Common sources of atmospheric POC and of SOC precursors are vehicular exhaust, industrial emissions, biogenic emissions, and biomass burning (Millet et al., 2005; Saarikoski et al., 2008; Genberg et al., 2011; Hu et al., 2012; Vodička et al., 2013; Heal and Hammonds, 2014; Huang et al., 2014a, b). Except near strong emission sources, secondary organic aerosol is the main contributor to the total organic aerosol mass concentration, frequently accounting for 72±21% (Zhang et al. 2007; Jimenez et al., 2009).

Trace metals account for only a small fraction of PM mass concentrations, but they can adversely impact human health (e.g., Lippmann et al., 2006; Hong et al., 2010). As some emission sources release specific trace elements, these elements can serve as useful source markers in PM source apportionment studies (Lee et al., 2011; Peltier and Lippmann, 2010; Han et al., 2005; Harrison et al., 2012; Karanasiou et al., 2009; Ondov et al., 2006; Querol et al., 2007; Viana et al., 2008; Yu et al., 2013). Elemental enrichments can also be used to roughly differentiate natural and anthropogenic sources (Khodeir et al., 2012; Rushdi, et al., 2013). Relative abundances of crustal elements can help identify the sources of suspended dust, as these abundances are known to be different for different dust source regions (Engelbrecht et al., 2009).

In this study, we report measurements of ambient particulate matter in Riyadh, the capital of Saudi Arabia. In the Middle East, dust / re-suspended dust is the major source of PM$_{10}$



(Ginoux, et al., 2012; Givehchi, et al., 2013); however, contributions from anthropogenic sources to PM mass concentrations are significant (>82% of total $PM_{10}$ mass, Al-Dabbous
and Kumar, 2015; >50% of $PM_{10}$ Tsiouri et al., 2015). Tsiouri et al. (2015) summarized the major sources of $PM_{10}$ in ambient air in the Middle East as oil combustion, re-
suspended soil, road traffic, crustal dust, and marine aerosol; significant sources of $PM_{2.5}$ were oil combustion in power plants, re-suspended soil, sand dust, and road traffic.
Carbonaceous particles were estimated to account for 50-60% of $PM_{2.5}$ in cities in Palestine, Jordan, and Israel (Abdeen et al., 2014). Not surprisingly, since oil production
and processing is widespread across the Middle East, heavy oil combustion was estimated to contribute 69% to $PM_{2.5}$ mass and 18% to $PM_{10}$ in in Jeddah, Saudi Arabia
(Khodeir et al., 2012). Air quality in Riyadh reflects not only the impact of local and regional dust and regional oil extraction and refining, but also significant local sources
that include a heavy traffic load and multiple industries. We focus here on identifying the major sources of $PM_{2.5}$ carbonaceous aerosol in Riyadh to provide a basis for formulating
air pollutant mitigation strategies.

## 2. Methodology

2.1 Sampling sites and data collection

Riyadh and its environs were divided into 16 12 km × 12 km sampling cells as shown in

Fig. 1. Sampling locations within each cell were carefully chosen to best represent the mix of land use and other characteristics of the cell.  From April through September, 2012,
an in-situ semi-continuous OC/EC analyzer (Sunset Laboratory Inc., Model-4), installed in a mobile laboratory, moved from cell to cell and measured hourly EC and OC, with
some interruptions due to instrument maintenance or holidays. The sampling strategy is documented in Table S1. In this instrument, volatile gases are removed from the samples
by carbon denuders prior to collection. Airborne particles smaller than 2.5 µm are then collected on quartz fiber filters at a flow rate of 8 l min$^{-1}$. Upon completion of a preset
sampling duration, all carbon that has been accumulated on the filter is removed by heating the sample in multiple increasing temperature steps, first in a completely oxygen-
free helium environment and then in a He/$O_2$ environment. The vaporized compounds flow through an oxidizer oven, are oxidized to carbon dioxide, and are detected via an
infrared analyzer. An external methane ($CH_4$) standard is injected at the end of every analysis and used to normalize the analytical result. Since in theory the quartz filter has
had all of the collected carbonaceous aerosol removed during each analysis cycle, the filter is re-used for multiple samples and changed only periodically.

A detailed description of the $PM_{10}$ sample collection and elemental analysis

methodologies can be found in Alharbi et al. (2015). In brief, sampling was conducted from the same mobile platform and concurrent with the OC/EC sampling. A $PM_{10}$ inlet
was used to sample ambient aerosol onto quartz microfiber filters over a 24 h period.



These samples were collected every three days and elemental analyses for Al, As, B, Ca, Cd, Co, Cr, Cu, Fe, K, Li, Mg, Mn, Mo, Na, Ni, Pb, Te, V, and Zn were performed by ICP-OES. NO and $NO_2$ ($NO_x$) were measured by chemiluminescence and $O_3$ was measured by UV photometer simultaneously using Signal Ambirak air quality monitoring system (Signal Ambitech Ltd, UK).

## 2.2 EC and OC re-split method

The Sunset semi-continuous EC/OC analyzer adopts the same thermal-optical analysis method for determination of OC and EC that is commonly applied to the offline analysis of filter samples. The OC and EC mass concentrations (as mass of C) are quantified by a calibrated non-dispersive infrared sensor (NDIR) signal that detects the evolved $CO_2$. Ideally, OC is defined as the carbon evolved under increasing temperature ramps conducted in an inert atmosphere (100% He) and EC is defined to be the subsequent carbon evolution in an oxidizing atmosphere (He/10% $O_2$ mixture). In the inert atmosphere, rather than simply volatilizing, a fraction of OC may be pyrolyzed due to insufficient oxygen, and this pyrolyzed OC may be evolved in the subsequent oxidizing atmosphere, appearing as EC. This fraction of OC is usually called pyrolyzed organic carbon (PyOC). To subtract PyOC from EC, laser transmittance or reflectance is deployed to monitor the variations in filter darkness; the transmittance or reflectance responds to the presence of EC throughout the analysis, but then drops when PyOC is formed and rises again as PyOC is evolved. The fraction of total assigned EC evolved in the oxidizing atmosphere before the laser signal returns to its initial value is believed to be due to PyOC, so in post-analysis the final EC is reported as the difference between the total carbon evolved in the oxidizing atmosphere and the PyOC. This methodology has been automated in the Sunset instrument. However, unusual EC and OC splits for a large number of samples were observed during the study period: (a) split points jumped to the end of the analysis because the laser response did not rebound to its initial value before the $CH_4$ calibration phase; or (b) split-points were located in the pre-oxygen position. These split point deviations may be ascribed to refractory residue on the filters: the laser correction factor supplied in the standard manufacturer software may not be applicable to the dusty environment of Riyadh (Polidori et al., 2006; Jung et al., 2011; Wang et al., 2012). Therefore, observed relationships between laser response and temperature in the $CH_4$ + $O_2$ injection calibration phase were used to develop a corrected split point. The correction methodology assumed that only refractory material was present on the filter in this phase, so that effects of this refractory material on the laser response to temperature variations could be isolated, corrected, and these corrections applied during the other analysis phases. A full description of the methodology is found in the Appendix. We note, however, that measurement artifacts from carbonates in dust may be present in this study, which would result in a high bias in the OC measurements. As noted in Karanasiou et al. (2011) and in the standard operating procedure (SOP) document published by the



Research Triangle Institute (RTI)
(https://www3.epa.gov/ttnamti1/files/ambient/pm25/spec/RTIIMPROVEACarbonAnalysis
SSLSOP.pdf), the evolution of carbonates from filter samples during thermal analysis can
occur over several carbon peaks. While it is preferred to use acid decomposition of
carbonates (on separate sample punches) to obtain the best quantification, Karanasiou
et al. (2011) demonstrated that the protocol used in this study completely evolves
carbonates in the OC fraction, and that manual integration to isolate the carbonate
concentration is possible but carries large uncertainty. Hence, we do not attempt to
separately quantify carbonate in this work.

2.3 SOC estimation by minimum R squared (MRS) method

The EC tracer method is widely used to estimate secondary organic carbon mass
concentrations, applying the following equations, which assume that EC has only
combustion sources:

$$POC = \left(\frac{OC}{EC}\right)_{pri} \times EC \qquad \text{Eqn 1}$$

$$SOC = OC_{total} - \left(\frac{OC}{EC}\right)_{pri} \times EC - b \qquad \text{Eqn 2}$$

where $(OC/EC)_{pri}$ is the OC/EC ratio in fresh combustion emissions, $b$ denotes non-
combustion-derived POC, and $OC_{total}$ and EC are ambient measurements. The key to
successful application of this method is to begin with an appropriate estimate of the
$(OC/EC)_{pri}$ ratio. Several approaches have been documented to determine $(OC/EC)_{pri}$.
Gray et al. (1986) directly adopted the ratios from emission inventories. Turpin and
Huntzicker (1995) used the measured OC/EC ratio when local emissions were dominant
in a certain location or over a specified period. Based on the expectation that co-emitted
POC and EC are well correlated, Lim and Turpin (2002) took the slope of OC against EC
using OC/EC ratio data for the lowest 5-10% values of that ratio. Millet et al. (2005)
proposed that a critical point where SOC is independent of EC should represent the
primary OC/EC ratio; the critical point is found by a minimum R-squared (MRS) method.
Assuming that non-combustion sources (i.e., the $b$ term in Eqn 2) are negligible, this
method can derive the most accurate primary OC/EC ratio, compared with previously-
proposed approaches (Wu and Yu, 2016). However, this method may underestimate the
SOC concentration if some SOC is associated with EC: co-emitted semi-volatile POC
could rapidly oxidize to low-volatility SOC and partition on the surface of EC. However,
given that accurate emission inventories are not available for Riyadh, we employed this
method in the absence of a priori knowledge of $(OC/EC)_{pri}$ to provide a conservative
estimate of the SOC concentration during our observational period.

The methodology for and applications of the MRS method were documented in Millet et
al., (2005), Hu et al., (2012), and Wu and Yu (2016). The non-combustion source ($b$ term)




is assumed to be zero. A series of coefficients of determination ($R^2$) between EC and
SOC calculated by Eqns 1 and 2, varying $(OC/EC)_{pri}$ from 0 to 10 using steps of 0.01 in
the ratio, is generated. At low $(OC/EC)_{pri}$ ratio, a significant portion of the estimated SOC
still belongs to POC. At high $(OC/EC)_{pri}$ ratio, the term $(OC/EC)_{pri} \times EC$ largely exceeds
$OC_{total}$ and becomes dominant. At the correct ratio, all the POC has been removed and
$R^2$ of SOC and EC reaches a minimum. This ratio is then used to estimate SOC in all
samples.

### 2.4 Back trajectory analysis

To develop an understanding of potential regional influences on observed PM, we
calculated 24-hr back trajectories (BTs) every 3 hours during each sampling period using
the National Oceanic and Atmospheric Administration (NOAA) Hybrid Single-Particle
Lagrangian Integrated Trajectory (HYSPLIT; Stein, et al., 2015; Rolph, 2016). Trajectories
were initiated for a starting height of 500 m above ground level (AGL). Residence time
analysis (RTA), describing the probability of air mass origins, was also performed
(Ashbaugh et al., 1985). The probability ($P_{ij}$), representing the residence time of a
randomly selected air mass in the $ij_{th}$ cell during the observational period, can be
calculated as follows.

$$P_{ij} \cong \frac{n_{ij}}{N} \qquad\qquad\qquad \text{Eqn 3}$$

where $n_{ij}$ is the number of trajectory segment endpoints that fell in the $ij_{th}$ cell and N is the
total number of endpoints.

Concentration weighted trajectory analysis (CWT) is another effective tool combined with
back trajectory data and pollutant concentration to trace the source origin for certain
species. The calculation formula is as follows.

$$C_{ij} = \frac{1}{\sum_{i=1}^{M} \tau_{ijl}} \sum_{i=1}^{M} C_i \tau_{ijl} \qquad\qquad \text{Eqn 4}$$

where $C_{ij}$ is the average weighted concentration in the grid cell (i, j), $C_i$ is the measured
species concentration, $\tau_{ijl}$ is the number of trajectory endpoints in the grid cell (i, j) and M
is the number of samples that have trajectory endpoints in the grid cell (i, j).

### 2.5 Positive matrix factorization (PMF) analysis

Positive matrix factorization (PMF) has been successfully applied to aerosol composition
data to suggest sources impacting the sampling site (Reff et al., 2007 and Viana et al.,
2008). We aligned daily-average OC and EC with concurrent averaged measurements of
metal concentrations in the $PM_{10}$ fraction (Al, Ca, Cu, Fe, K, Mg, Mn, Na, Ni, Pb and V)
and prepared a matrix with size of 35 ×13 for input to the USEPA PMF, version 5.0



(https://www.epa.gov/air-research/positive-matrix-factorization-model-environmental-data-analyses). Data points with "ND" were replaced by ½ of the detection limit and the corresponding uncertainties were assigned as 5/6 of the detection limit. The uncertainties
for all other data were calculated as $s_{ij} + DL_{ij}/3$, where $s_{ij}$ represents the analytical uncertainty for species i in the sample j and $DL_{ij}$ represents the detection limit (Polissar
et al., 1998; Reff et al., 2007). In this study, the analytical uncertainty was assumed to be 5% of the corresponding concentration for metal species. Uncertainties for the EC and
OC data were not reported. Norris et al. (2014) suggested that, for such cases, the initial uncertainties be set to a proportion of the concentration. The uncertainties for OC and EC
were therefore calculated as 10% of the corresponding concentrations for this study.

## 3. Results and discussion

### 3.1 Overview of EC and OC concentrations

Fig. S1.a shows the time series of OC and EC concentrations during the study period and

denotes the corresponding sampling cells in which the measurements were obtained. Average OC and EC concentrations during the observational period were 4.8 ± 4.4 and
2.1 ± 2.5 µg C m$^{-3}$, respectively (we will use µg m$^{-3}$ for OC and EC hereafter when referring to µg C m$^{-3}$). Table 1 presents some comparative values of measured EC and OC
concentrations in PM$_{2.5}$ in urban areas world-wide. The average concentrations in this work for both EC and OC were remarkably consistent with those reported by Abdeen et
al. (2014) for 11 Middle Eastern sampling sites. The average OC concentrations were also comparable to those reported for suburban Hong Kong (4.7 µg m$^{-3}$, Huang et al.,
2014b), but lower than those reported for Veneto, Italy (5.5 µg m$^{-3}$, Khan et al., 2016), Athens, Greece (6.8 µg m$^{-3}$, Grivas et al., 2012), urban Hong Kong (10.1 µg m$^{-3}$, Ho et
al., 2006), Delhi, Indian (16.5 ± 6.6 µg m$^{-3}$, Satsangi et al., 2012), and Beijing, China (18.2 ± 13.8 µg m$^{-3}$, Zhao et al., 2013), reflective of the different mix of sources and different
photochemical environments. EC concentrations were similar to those in Athens, Greece, higher than those reported for Veneto, Italy and suburban Hong Kong, and lower than all
other measurements shown in Table 1.

The Riyadh sampling site characteristics and the corresponding average OC and EC

concentrations in each grid cell are summarized in Table S1. Results of a one-sided t-test (p<0.001) on OC and EC data from industrial and residential sites suggest a significant
difference in carbonaceous aerosol concentrations between the two site types: OC mass concentrations in industrial sites were 1.4 times those in the residential sites, and EC
mass concentrations were 1.7 times higher (Fig. 2). The mean OC/EC ratio was lower in the industrial sites (3.1) than in residential sites (6.0), suggesting the importance of POC
emissions in industrial regions and a larger SOC contribution in residential areas. We also divided Riyadh into four quadrants to investigate the spatial variation of OC and EC across
the city. Fig. 3 shows that OC and EC concentrations were higher in the eastern quadrants.



Fig. 4 shows the results of the RTA, demonstrating that air masses arriving in Riyadh
were mainly from within Saudi Arabia and from the south / southwest of the city in April
and May, and from the north / northeast from June to September, extending to the west
coast of the Persian Gulf. These two dominant wind directions have been used to stratify
data in Fig. S1b, which shows that the average OC concentration increased from 3.8 to
5.3 µg m$^{-3}$ and EC from 1.1 to 2.7 µg m$^{-3}$ when the air mass source region shifted from
south/southeast to north/northeast, respectively.

**3.2 Weekend effect in OC and EC concentrations**

A "weekend effect" in concentrations of traffic-derived PM has been noted in previous
studies (e.g. Grivas et al., 2012; Bae et al., 2004; Moteballi et al., 2003; Limand Turpin,
2002; Jeong et al., 2004; Lough et al., 2006). To investigate whether a weekend effect
can be discerned in the Riyadh dataset, two-sample t-tests assuming unequal variances
were performed for hourly EC and OC samples, grouped according to whether they were
obtained on weekdays (Sunday to Thursday) or on weekends (Friday and Saturday). The
test indicated a statistically significant difference (29% lower on weekends) in EC
concentrations between weekday and weekend, but no significant difference in OC (p <
0.001 with a 95% confidence level), as shown in Fig. 5. NO$_x$ concentrations were also
reduced during weekends, by 48% compared to weekdays (Fig. S2). This reduction may
be ascribed to the decrease in vehicular activities and industrial activities during the
weekend. Since OC concentrations had no significant weekday-weekend variation, the
increase in OC/EC ratio during the weekend likely indicates the importance of regional
photochemical sources of SOC, although decreased NO$_x$ emissions on weekends may
promote more efficient photochemical processing of local SOC precursors (Gentner et al.,
2012).

**3.3 Diurnal variation of OC and EC**

Fig. 6 shows the diurnal variations in OC and EC mass concentrations. OC and EC
concentrations peaked from 6-9 am and were elevated during nighttime (after 1600 pm).
NO$_x$ also shows a similar diurnal pattern. The morning peak coincides with traffic rush
hours. The elevation of OC, EC and NO$_x$ at night after 1600 pm may be attributed to the
accumulation of pollutants in the shallower nocturnal boundary layer. Average OC/EC
ratios showed no obvious trends; however, the median OC/EC ratio decreased slightly
over the time period when OC and EC concentrations built up, probably due to the
increased contributions from primary emissions. The average OC/EC ratio had a peak
around 14:00 pm, corresponding with peak concentrations of O$_3$, suggestive of secondary
aerosol formation.

**3.4 SOC estimation**





Fig. 7 shows the determination of $(OC/EC)_{pri}$ using the minimum R squared method (MRS). The value of this ratio derived in this study is 1.01, which occurred at the 14$^{th}$ percentile in the observed OC/EC ratios. In the compilation of $PM_{2.5}$ OC and EC emission profiles presented by Chow et al. (2011), the $(OC/EC)_{pri}$ for oil combustion is documented to range from 0.2 to 2.5 with an average of 1.0±0.2, 0.9 to 8.1 with an average of 3.4±2.2 for gasoline emissions, and 0.2 to 2.7 with an average of 1.0±0.8 for diesel emissions. Our estimate is within with these ranges and is closer to the averages for oil combustion and diesel emissions, consistent with expected important contributions from these sources to $PM_{2.5}$ carbonaceous aerosol in Riyadh. Using our MRS-derived $(OC/EC)_{pri}$ in equations (1) and (2), we computed average POC and SOC concentrations of 2.0±2.4 and 2.8±3.4 µg m$^{-3}$, respectively, suggesting that POC and SOC contributions to $PM_{2.5}$ were comparable during our study. The average POC and SOC concentrations were 1.0±1.0 and 2.7±4.0 µg m$^{-3}$, respectively, when transport was from the south/southwest. POC increased to 2.5±2.7 µg m$^{-3}$ and SOC was almost unchanged when the direction of transport was from the north / northeast. Variability in OC was thus mainly due to variability in POC. The sampling locations were in cells classified as being in the outskirts of the city when south/southwesterly transport was prevalent, but included both outskirts and in-city grids when north/northeasterly transport was prevalent. The increase in POC during northerly transport regimes could not therefore be attributed solely to the influence of local primary emissions, since transport of POC from outside Riyadh was also possible.

The diurnal variation of SOC (Fig. S3) showed a small peak of SOC concentration in the morning from 7-9 am, lagging behind the POC and EC morning peaks by about two hours; this result is not unexpected since photochemical production of SOC will require time for reactions to proceed once precursors have accumulated in the atmosphere. A second small peak in SOC concentration occurred at 14:00 pm, concurrent with ozone formation (Fig. S4) and consistent with the variation in OC/EC ratios discussed in Section 3.3. The estimated POC was 2.2±2.5 µg m$^{-3}$ on weekdays and decreased to 1.5±1.9 µg m$^{-3}$ on weekends. The estimated SOC was 2.6±2.9 µg m$^{-3}$ on weekdays and increased by 23% to 3.2±4.5 µg m$^{-3}$ on weekends. With regards to spatial variation, POC and SOC were 3.5±2.7 and 3.2±2.9 µg m$^{-3}$ in the industrial sites, 2.1±2.6 and 2.6±3.0 µg m$^{-3}$ in the residential sites, and 1.1±1.1 and 2.8±4.1µg m$^{-3}$ in the outskirts sites, respectively. SOC concentrations were 2.5 times those of POC in the outskirts sites, an expected result since these latter sites are farther removed from the sources of primary emissions within the city proper.

## 3.5 Possible sources of $PM_{2.5}$ carbonaceous aerosols

### 3.5.1 Correlation between OC, EC and other elemental species

As a first step in seeking signatures of sources of carbonaceous aerosol in Riyadh, we conducted an analysis of the correlations between OC or EC and measured elemental



species. We note that OC and EC were measured in the PM$_{2.5}$ fraction, while elemental
species concentrations were obtained for the PM$_{10}$ fraction, which also includes the PM$_{2.5}$.
OC and EC were poorly correlated with K, which we interpreted as indicating a negligible
influence of biomass burning on PM. Al, Fe, Mg, Mn, and Ca are found in crustal soils
and in PM samples of windblown dust. EC did not correlate well with these species (R$^2$ <
0.35; not shown). However, OC had a relatively strong correlation with Ca (R$^2$ of 0.63)
(Fig. 8) but, similar to EC, a poor correlation with other dust species (not shown). These
findings indicate that OC may have shared a source with Ca, but this source is not likely
to be associated with windblown dust. The correlation between SOC and Ca was stronger
than that between POC and Ca (Fig. S5). The thermo-optical method may have measured
CaCO$_3$ as OC, and the subsequent estimates of SOC separated two sources, one
associated with combustion and EC ("primary"), and another associated with CaCO$_3$ (and
mis-labeled "secondary"). Concentrations of Al and of other metals (Fe, K, Mg and Mn)
were strongly correlated (R$^2$>0.9), supporting their common dust origin (Fig. 8). The
correlation between Ca and other dust species, however, shows two divergent regimes,
suggestive of an additional Ca-containing source besides dust. Therefore, understanding
the sources of Ca becomes a prerequisite in understanding the sources of OC.

The enrichment factor (EF) is a practical and convenient tool to differentiate natural and
anthropogenic sources of metal species (Khodeir et al., 2012; Rushdi, et al., 2013). The
EF can be calculated using the following equation (Taylor, 1964):

$$EF = \frac{\left(X / C_{ref}\right)_{air}}{\left(X / C_{ref}\right)_{source}}$$
Eqn 5

where X is the measured metal concentration, and $C_{ref}$ is the concentration of the
reference metal species. The equation compares the ambient elemental abundance of
two species with their source abundance. An EF less than 10 suggests that the sample
may come from a natural crustal source and an EF value > 10 indicates possible
anthropogenic influence (Biegalski et al., 1998). Al, Fe, and K were all used as reference
species to test for robustness of the findings. Fig. S6 shows that, for all three reference
species, the EFs for Al, Fe, K, Mn, Mg, Na and V were calculated to be less than 10,
suggesting a dominant crustal type origin. Ni, Zn, Cr, Co, Pb, Li, B, As, Mo, Cd, and Te
were calculated to be larger than 10, suggestive of the influence of anthropogenic
emissions, e.g. traffic emissions, fossil fuel combustion and non-ferrous metal industries.
The EF for Ca was calculated to be ~10, consistent with the idea that it may have two
sources in Riyadh, one natural and one anthropogenic.

Cement kilns are documented to be important sources of elemental Ca in the atmospheric
aerosol (Zhang et al., 2014). Chow et al. (2004) noted an important contribution of PM$_{2.5}$
POC from cement factories. Hence, contributions from cement production sources may



have led to the good correlation between OC and Ca at the receptor sites. In the Middle East, another possible anthropogenic source for Ca is from the desalting and demetalization of crude oil in refineries (Wu et al., 2014); refineries are certainly
contributing to the observed OC in Riyadh. A third possibility is that the Ca is crustal in origin, but from a different source region than most of the other sampled dust. Ca
enrichment in dusts may vary across the Middle East region (Coz et al., 2010), and thus the correlation between Ca and other crustal species could diverge depending upon the
dust source region. Regardless of dust source region, during transport to Riyadh, as ambient SOC precursors are oxidized, the products may be partitioned to particle
surfaces, resulting in simultaneous transport of Ca and OC. Finally, it is important to note that a correlation between Ca and OC may occur if calcium carbonate is being sampled
and the carbonate detected as OC in the thermal analysis protocol, as mentioned in the Methods section above. While it is not possible to definitively distinguish between these
various possibilities based only on EF, the large dust loadings that were present in nearly all samples suggest that this latter explanation could play a significant role in producing
the observed Ca-OC correlations.

### 3.5.2 CWT analysis for Ca/Al ratio, Pb, OC and EC

We used CWT analysis to identify possible source origins for the observed highest values of Ca/Al ratio, Pb, OC and EC (Fig. 9). The CWT plot for the Ca/Al ratio shows that, when
this ratio was high in Riyadh PM samples, air masses were most likely to have passed over regions along the western shoreline of the Persian Gulf, and in particular, the highest
ratio was found for air masses passing over the site of a large Saudi Arabian cement plant (Fig. S7). This transport pathway is thus consistent with the idea that refineries and
cement plants may represent anthropogenic sources of Ca. CWT analysis of Pb shows that high observed concentrations in Riyadh aerosol were associated with transport from
Iraq, consistent with the continued usage of leaded fuel in that country (Shaik et al., 2014). $PM_{10}$ Pb concentrations were $0.035\pm0.088$ µg m$^{-3}$ in this study, lower than measurements
reported for eastern China (0.05 to 0.5 µg m$^{-3}$, Li et al., 2010) and the greater Cairo area (0.3 µg m$^{-3}$, Safar and Labib, 2010), both locations for which leaded fuel has been phased
out of usage`, and lower than the U.S ambient concentration standard for lead (0.15 µg m$^{-3}$ on a 3 month rolling basis; U.S. EPA, 2006). The comparison shows that although Pb
may have multiple potential sources in Riyadh, the concentration levels are still below those of concern for human health. Industrial emissions along the Saudi Arabian coast
may also contribute some Pb to the measured aerosol. While high OC concentrations were associated with transport from a similar region of the Persian Gulf as was high Pb,
the high-concentration source region extended further north, encompassing areas with oil fields and refineries and the Baghdad urban region (Fig. S8). Finally, the CWT plots
for OC and EC are similar, suggesting their highest concentrations may be attributed to similar sources, i.e., refineries, cement factories and urban pollution.



### 3.6 PMF analysis

Three- to five-factor solutions were tested in the PMF model; the three-factor solution was found to have the best solution characteristics (Fig.10). Most of the OC (77%) and EC (90%) together with fractions of the crustal elements appear in the first factor. We note that 54% of Ca is loaded in this factor, as expected since OC was found to be correlated with Ca. No significant crude oil tracers (Ni and V) appear in the factor, indicating that this factor is not related to oil combustion (Ganor et al., 1988). The CWT analysis suggested that high OC and EC coming from the shoreline of the Persian Gulf may be associated with industrial emissions, including refineries, gas flares in oil fields, and cement production. However, we cannot rule out potential contributions to this factor from local vehicular emissions. Therefore, this factor is identified as a mixed source: cement industries / gas flares / local vehicles.

A key signature in the second factor is the significant loading of Pb (98%); it also includes some dust species. While leaded fuels have been phased out in Saudi Arabia, as mentioned above, they are still in use in Iraq; further, deposition of lead to soils and resuspension is a documented exposure pathway (Laidlaw and Filippelli, 2008). CWT analysis also supports a source origin of Pb from Iraq (Fig. 9). Hence Pb may serve as a regional transport tracer in this PMF analysis. However, Pb can also be contained in other industrial emissions, including cement manufacturing in the city. The second factor is thus identified as leaded fuel combustion from long range transport / industrial emissions.

The third factor contains almost all of the V and a large fraction of Ni (>60%), as well as some crustal elements and OC. V and Ni and their ratios have been suggested as markers of emissions from oil fired power plants (Ganor et al., 1988). Barwise (1990) found that the highest V/Ni ratios (>1) among oil samples that they characterized were associated with Abu Dhabi and Suez oils, as contrasted with samples from the North Sea, China, Indonesia, and Australia, reflecting geological differences. The ratio of V/Ni in factor 3 is 3.5, consistent with the Arabian Gulf source of oil in this region. Dust species and some OC and EC are also associated with this factor, which we therefore identify as oil combustion.

Fig. 11 shows the source contribution to OC and EC from these three factors in individual samples. On average, the OC concentration was dominated by the mixed source (factor 1) (3.8 µg m$^{-3}$, 77%), followed by leaded fuel from long range transport (0.8 µg m$^{-3}$, 27%) and oil combustion (0.3 µg m$^{-3}$, 6%). The contribution of the mixed source ranged from 37% in May (0.7 µg m$^{-3}$) to 97% in September (7.6 µg m$^{-3}$). The EC concentration was also mainly attributed to the mixed source (1.9 µg m$^{-3}$, 92%).

### 4. Conclusions



To our knowledge, this study represents the first reported long-term and spatially resolved hourly measurements of ambient OC and EC concentrations for Riyadh, Saudi Arabia, along with supporting measurements that enable a source apportionment of these important aerosol species. We found that OC and EC average concentrations were comparable to other reported measurements in Middle Eastern cities, and diurnal and weekly variations indicated a clear influence from local emissions. However, OC and EC concentrations varied with air mass source origin, indicative of not only variations across Riyadh and its outskirts, but also of the influence of regional sources on carbonaceous aerosol concentrations. Due to the limited sample size, this study could not separately quantify the local and regional source contributions for OC and EC. About half of the measured OC was attributed to secondary formation, and positive matrix factorization suggested that EC and OC were mainly attributed to a mixed source category comprising cement industries, gas flaring activities, and local vehicles.

Measurement of OC and EC via the online thermo-optical technique was found to be challenging in the dusty environment encountered year-round in Riyadh. Our dataset required correction via a hand analysis, as reported in the supplementary materials, as the automated split method implemented by the manufacturer frequently failed for our samples. The lack of a separate independent carbonate analysis, however, means that our reported OC concentrations may be biased high, as also suggested by the strong correlation between OC and Ca. However, the correlation between OC and Ca may also suggest co-emission of OC and its precursors with metal Ca from desalting and demetalization processes in refineries; co-emission of Ca and OC from cement plants; or condensation of OC on Ca-rich dust during long-range transport. In future studies of ambient aerosol OC concentrations in dusty environments via online thermo-optical techniques, additional observations or different measurement protocols are needed to separate the contributions of carbonates to the measured OC and EC concentrations. With such added information, the implied sources of Ca and OC can be further investigated and their potential contributions to observed OC quantified.

**Appendix A: Correction method for OC/EC splits in data from the Sunset semi-continuous analyzer**

Laser response and temperature for individual blanks were well correlated, suggesting that the influence of temperature on laser response may indirectly affect the EC/OC split points (Figure A.1). This phenomenon has been pointed out previously, and Versions RT-Calc 114 and newer of the Sunset instrument analysis software introduced a laser correction factor to counteract the influence of temperature on the laser signal. This correction factor is calculated in each cycle from the variation in the laser signal when the analysis enters the methane calculation stage (Jung et al., 2011). However, it was



obvious that this correction approach did not work well for the Riyadh samples, since many returned EC/TC=0, the case when the initial reflectance is not recovered in the
analysis. A revised method of finding the point of return to the original laser signal, and thus determining the POC and EC contributions, was therefore proposed for this study
and used to correct the dataset.

The relationship for the Riyadh samples between laser response and temperature
during the calibration phase of the $CH_4 + O_2$ injection was used to develop a corrected split point, assuming that only refractory material is present in this phase, and the effects
of this refractory material on the laser response to temperature variations could be isolated and then applied during the other analysis phases. A correlation between laser
response and temperature in the calibration phase was derived using linear and quadratic functions. The derived parameters from the two functions were applied in the following
equations to recompute a corrected laser signal for each analysis, instead of the laser correction factor automatically generated by the Sunset program:

$$Signal_{new} = Signal_{original} + a \ (Temp^2_{initial} - Temp^2_{original})$$

$$+ \ b \ (Temp_{initial} - Temp_{original}) \hspace{3cm} \text{(A.1)}$$

$$Signal_{new} = Signal_{original} + c \ (Temp_{initial} - Temp_{original}) \hspace{2cm} \text{(A.2)}$$
where $Signal_{original}$ represents the original laser signal, $Signal_{new}$ represents the signal after correction to the initial temperature, $Temp_{initial}$ represents the temperature at the
initial condition when each analysis begins, and $Temp_{original}$ represents the original temperature for each analysis; a and b in Eq. (A.1) are derived from the quadratic
equation for each analysis, and c in Eq. (A.2) was derived from a linear fit.

Since refractory residues accumulated on the filter during the measurement period,
the derived correlation between laser response and temperature varied sample by sample. The equations to derive the corrected laser signal were, therefore, applied individually to
each sample. In the blank sample, the quadratic-function-generated laser signal was smoother than the linear-function-generated one, especially during the calibration phase
of the $CH_4 + O_2$ injection (Figure A.2a). The relationship between temperature and laser signal for the newly replaced filter tended to be closer to linear, while the signal for the
aged filter with residue accumulation showed a better fit using a quadratic equation. A quadratic equation was therefore selected to correct the laser signal for the entire dataset.
The new split points were then set to where the corrected laser signal rebounded to its





value just before OC pyrolyzed and the laser signal decreased due to pyrolyzed organic
carbon formation. The method worked for both incorrect split-point cases, bringing the
split point back to the He + $O_2$ phase as expected and leading to more reasonable EC/OC
split points, i.e., neither at the end of the analysis nor in the pre-oxygen analysis phase.

### Acknowledgment

The authors gratefully acknowledge the financial support of King Abdulaziz City for
Science and Technology (KACST) under grant number 32-594 and the NOAA Air
Resources Laboratory (ARL) for the provision of the HYSPLIT transport and dispersion
model and/or READY website (http://www.ready.noaa.gov) used in this publication.

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



Table 1 Comparison of OC and EC (µg m$^{-3}$) measured in the different cities

| City | Duration | EC | OC | EC | OC | References |
|------|----------|-----|-----|-----|-----|------------|
| | | Conc. (µg m$^{-3}$) | | S.D. (µg m$^{-3}$) | | |
| Anthens, Greece | Jan to Aug, 2003 | 2.2 | 6.8 | | | Grivas et al., 2012 |
| Riesel, TX, US | May, 2011 to Aug, 2012 | 0.18 | 2.67 | 0.09 | 1.62 | Barrett and Sheesley, 2014 |
| Beijing, China | Selective days in four seasons from 2009 to 2010 | 6.3 | 18.2 | 2.9 | 13.8 | Zhao et al., 2013 |
| Urban, Hong Kong | Nov, 2000 to Feb, 2001 and Jun, 2001 to Aug, 2001 | 5.71 | 10.12 | 0.89 | 1.92 | Ho et al., 2006 |
| Suburban, Hong Kong | Mar, 2011 to Feb, 2012 | 0.86 | 4.70 | 0.53 | 2.87 | Huang et al., 2014b |
| Middle east (11 sampling sites in Palestine, Jordan and Israel) | Jan to Dec, 2007 | 2.1 | 5.3 | 2.2 | 4.0 | Abdeen, et al., 2014 |
| Veneto, Italy | Apr 2012 to Feb 2013 | 1.3 | 5.5 | | | Khan et al., 2016 |
| Delhi, India | Dec 20, 2012 to Feb 26, 2013 | 12.04 | 16.46 | 4.43 | 6.61 | Panda et al., 2016 |
| Riyadh, Saudi Arabia | Apr to Sep, 2012 | 2.13 | 4.76 | 2.52 | 4.4 | this study |





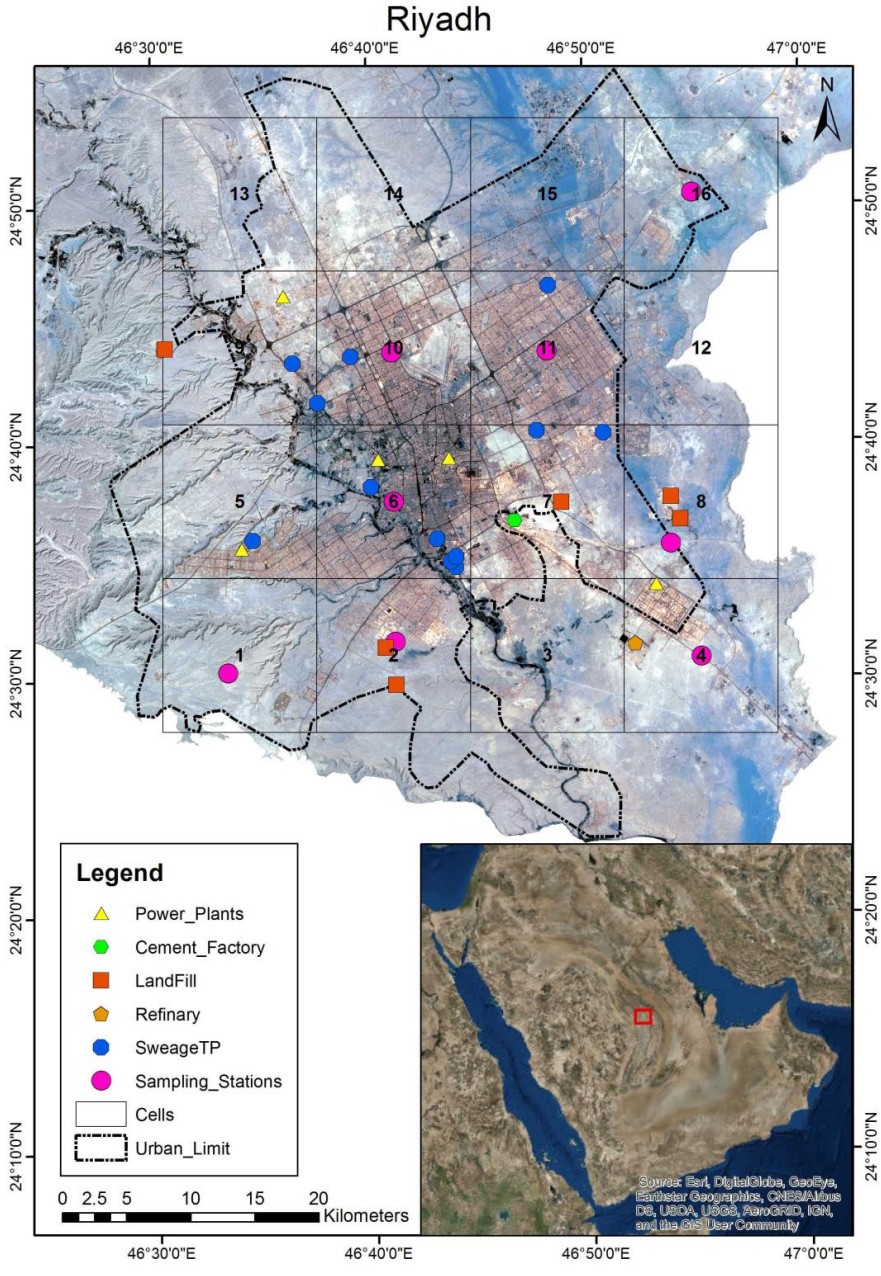

Figure 1 Image of Riyadh and immediate surroundings. Potential emission sources and 16 sampling locations are indicated. The characteristics of the sampling locations are listed in Table 1.



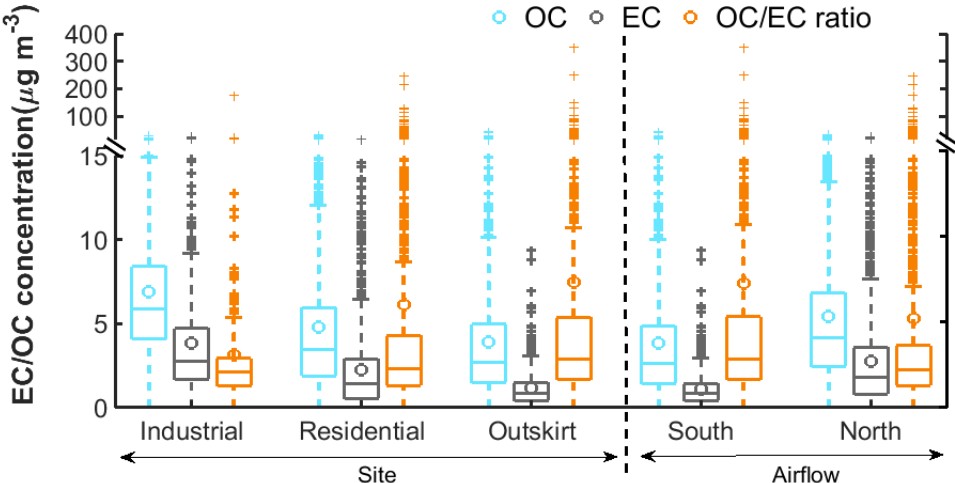

Figure 2 Observed OC and EC concentrations (μg m⁻³) separated by site types and air mass source region according Table 1 and Figure 1b. Box and whisker plots show median and quartile values; averages are shown as circles and outliers as crosses.

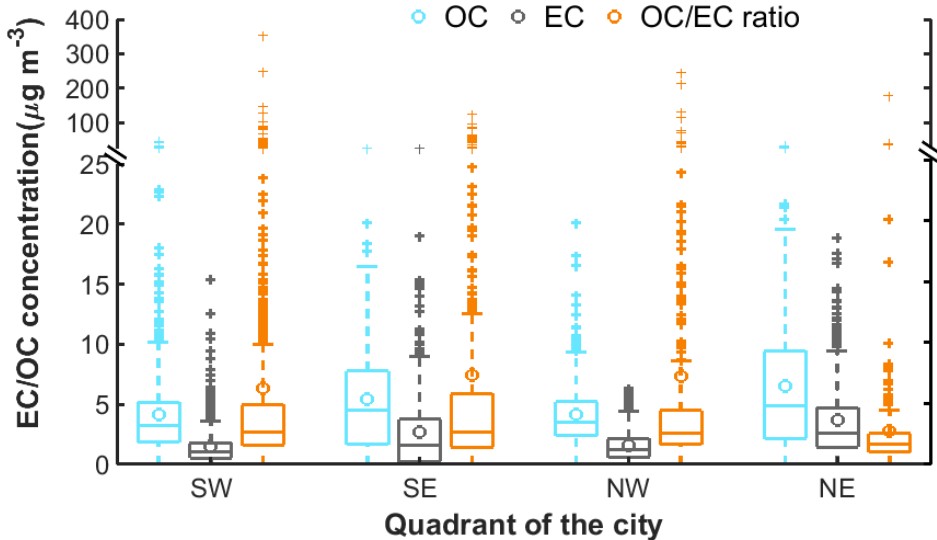

Figure 3 Spatial variation of OC and EC concentrations (μg m⁻³) and OC/EC ratios in each quadrant of Riyadh. SW represents southwest Riyadh and includes the sampling cells 1, 2, 5 and 6; SE represents southeast Riyadh and includes the cells 3, 4, 7, and 8; NW represents northwest Riyadh and includes the cells 9, 10, 13, and 14; and NE represents northeast Riyadh, and includes cells 11, 12, 15, and 16.





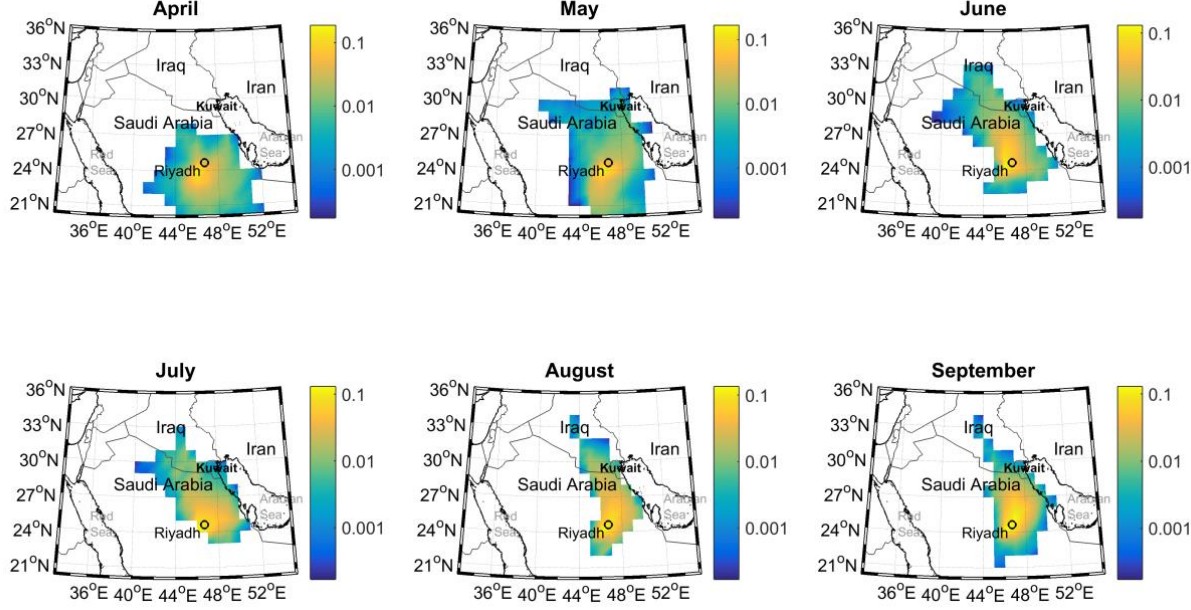

Figure 4 Back trajectory (24 h) residence time analysis of air masses arriving at Riyadh from April to September, 2011. Back trajectories were initiated from a starting height of 500 m above ground level. The color bar represents the normalized number count of the end points.





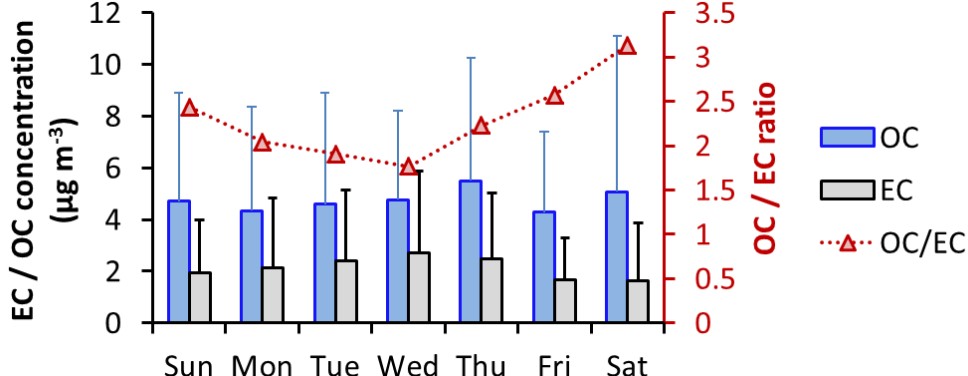

Figure 5 Day-of-week variation in OC ($\mu$g m$^{-3}$), EC ($\mu$g m$^{-3}$) and OC/EC ratio during the observational period.





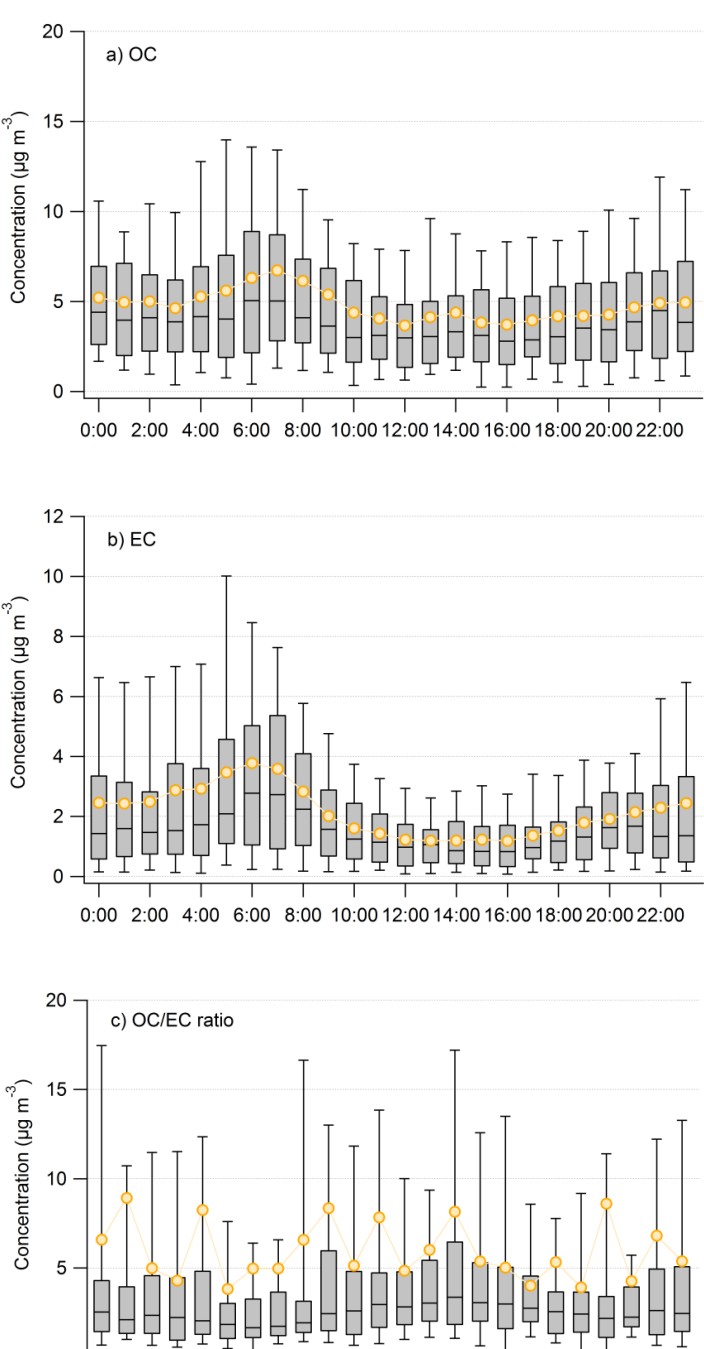

Figure 6 Diurnal variation of a) OC, b) EC and c) OC/EC ratio. Box and whisker plots show median and quartile values; averages are shown as orange circles.



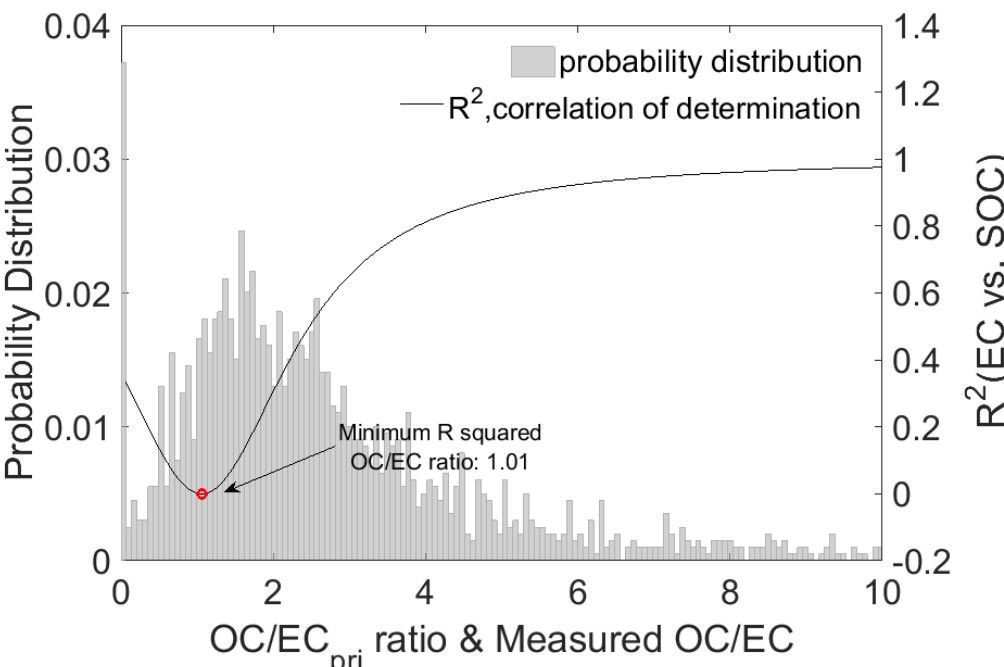

Figure 7 Determination of (OC/EC)$_{pri}$ using the minimum R squared method (MRS). The black curve is the coefficient of determination ($R^2$) between SOC and EC as a function of the assumed primary OC/EC ratio. The grey shaded area represents the probability distribution of the measured OC / EC ratios. The turning point (red circle) in the curve gives the best-fit primary emission ratio (OC/EC)$_{pri}$.





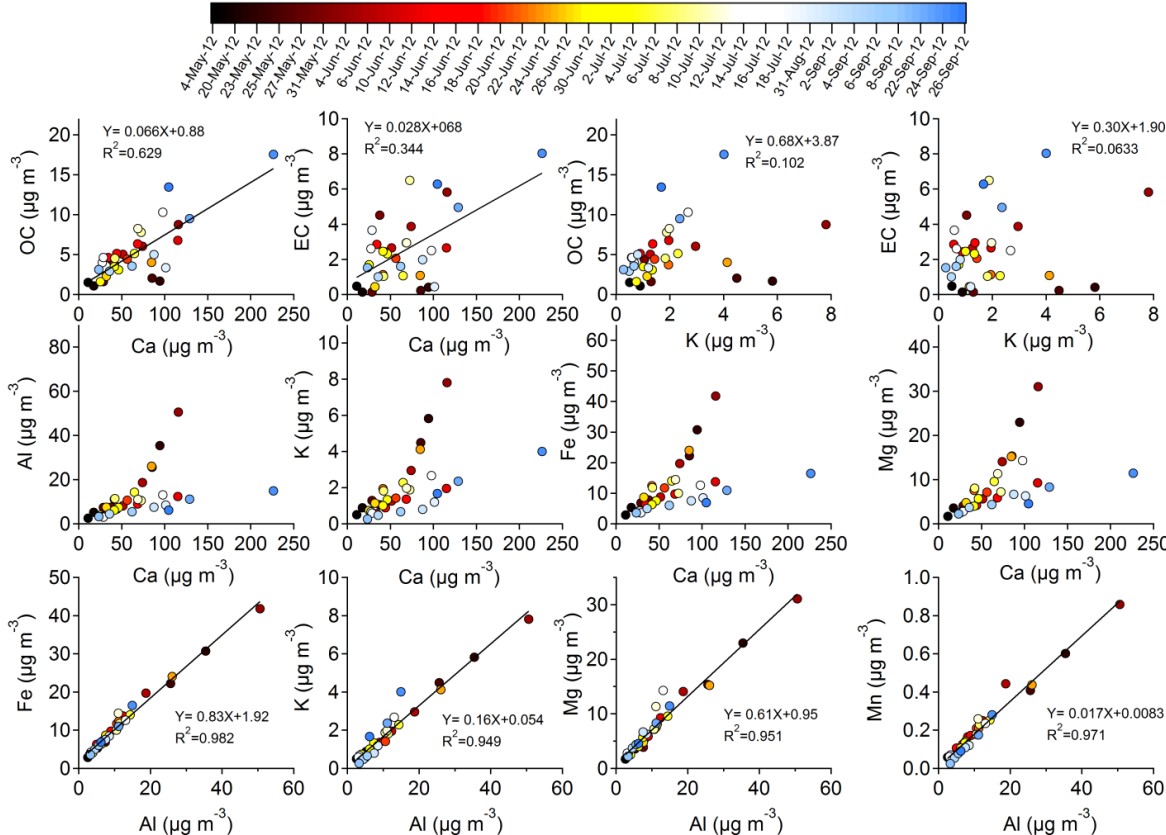

Figure 8 Correlation between dust species (Al, Fe, K, Mg, Mn and Ca), organic carbon (OC) and elemental carbon (EC) concentrations (µg m$^{-3}$). Color bar represents the corresponding sampling date.





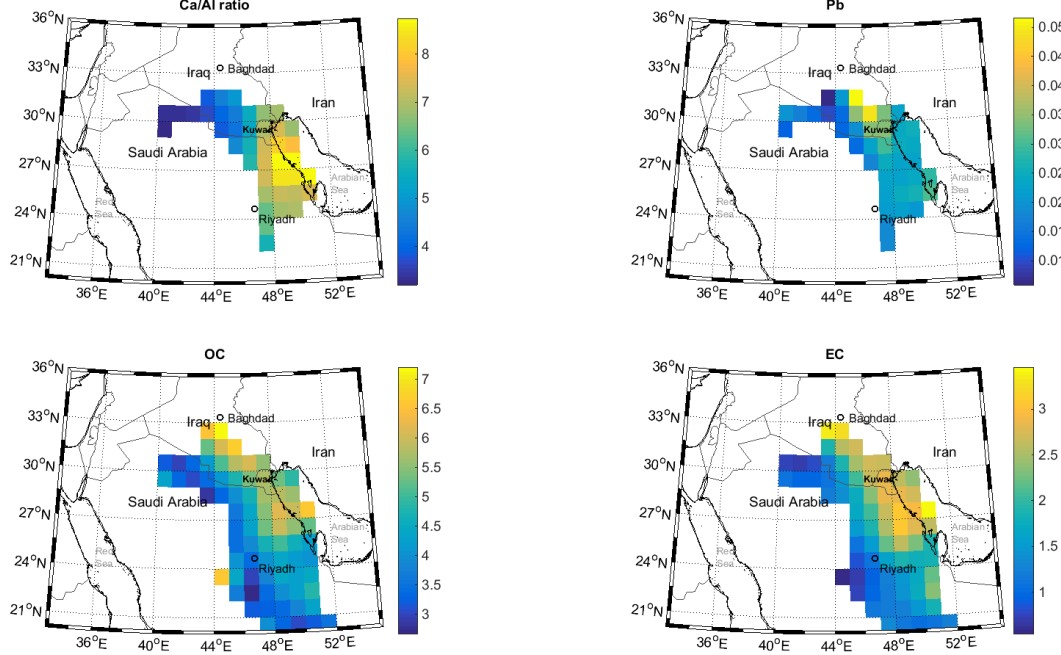

Figure 9 Concentration weighted trajectory analysis for indicated species, for 24-hr back trajectories with a starting height of 500 m. Color bars represent Ca/Al ratio, Pb concentrations (ng m$^{-3}$), OC concentrations (μg m$^{-3}$), and EC concentrations (μg m$^{-3}$).





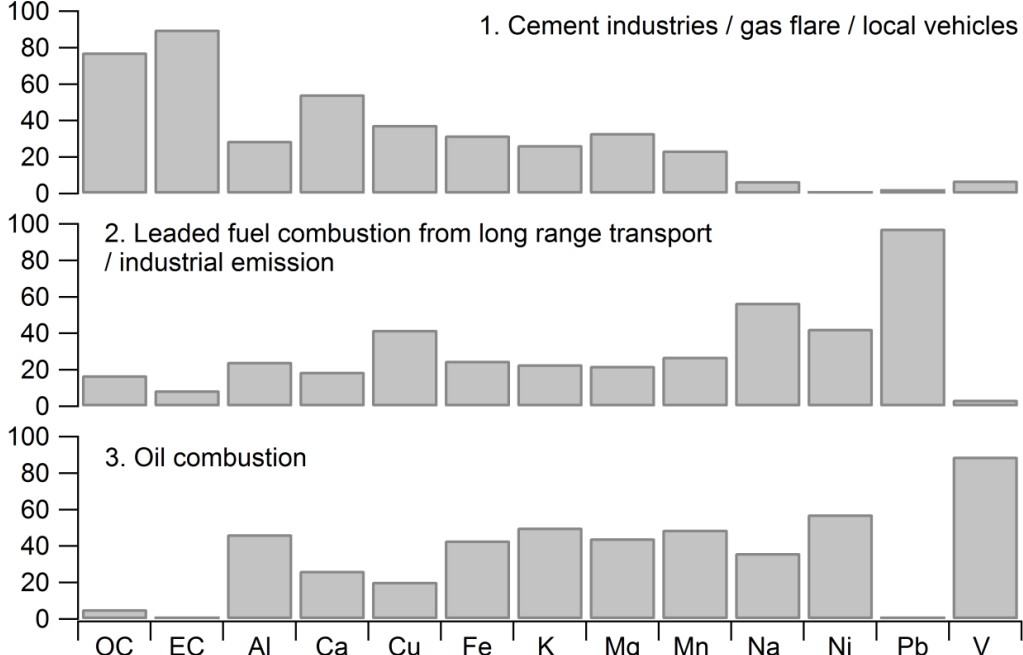

Figure 10 Source profile of PMF analysis of combined $PM_{2.5}$ OC and EC and $PM_{10}$ metals concentrations. The sum of the species for all the factors was normalized to unity.





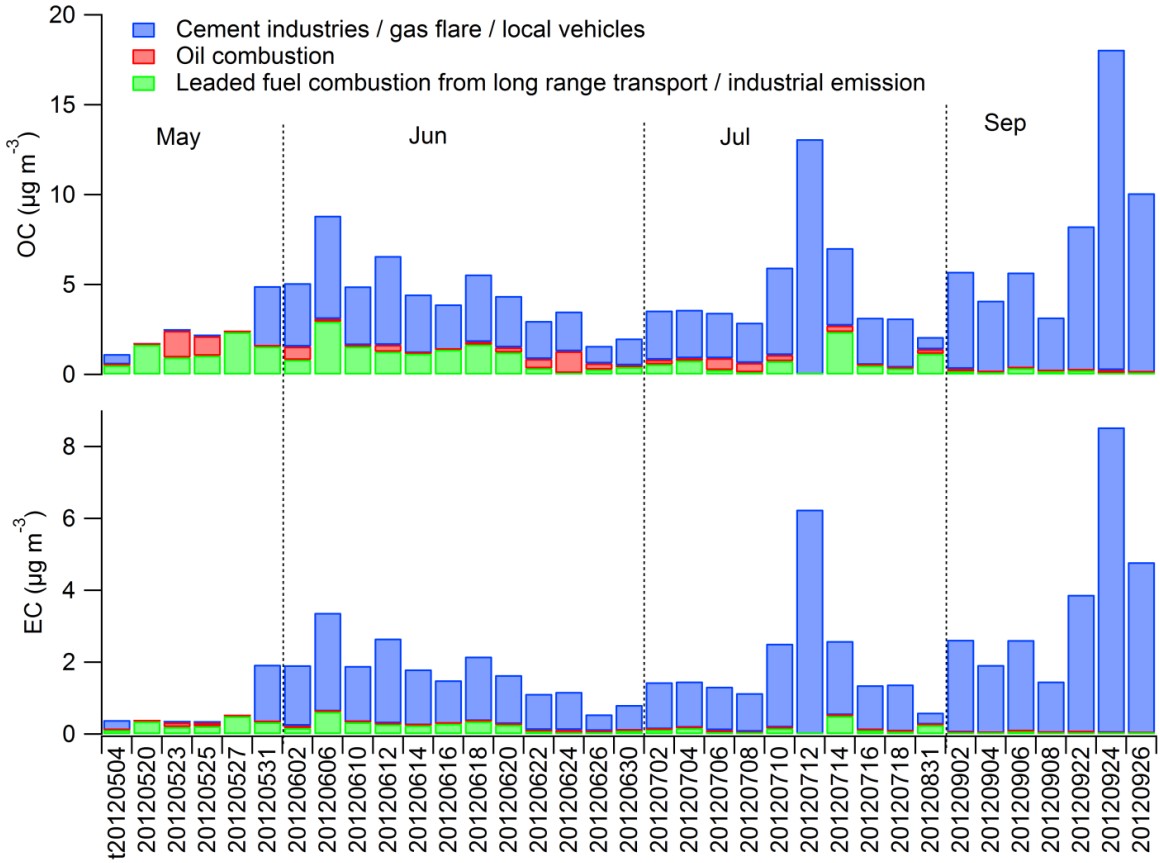

Figure 11 Source contributions to (a) OC and (b) EC (µg m$^{-3}$) from three sources, for each sample.





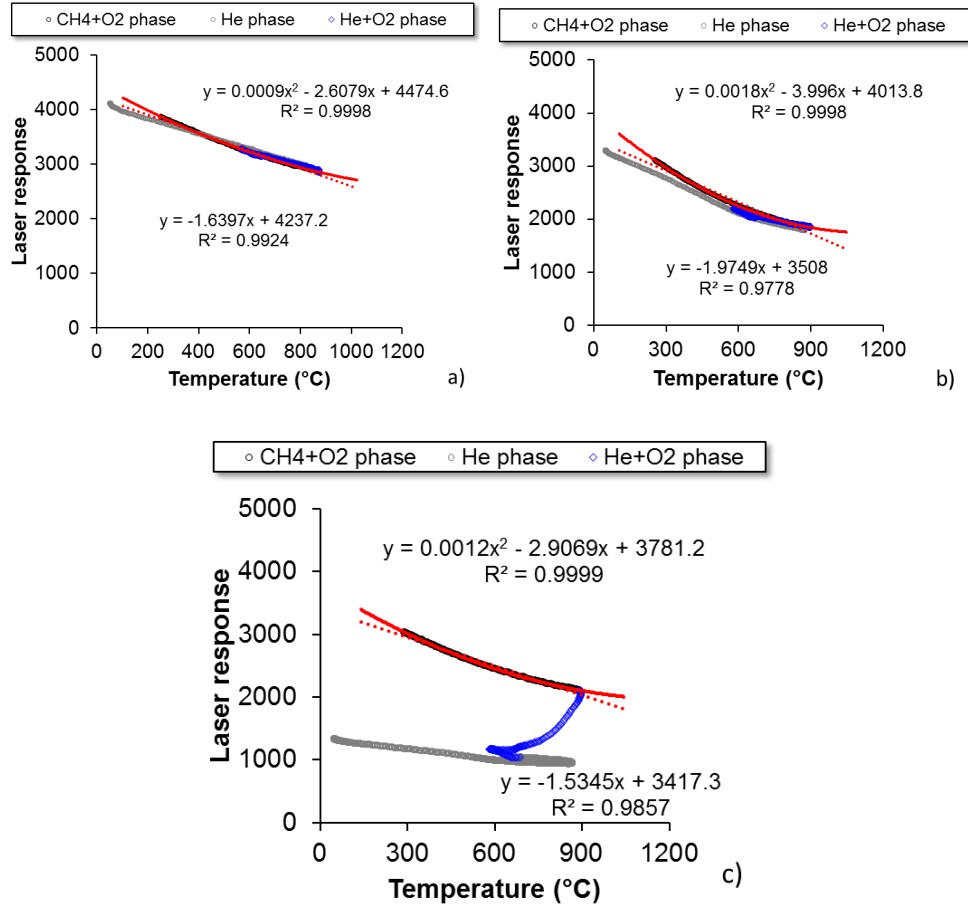

Figure A.1: Correlation between laser response and temperature (°C) for the three samples whose thermograms are shown in Figure A.2. (a) blank at 00:15 am, 20120706; (b) ambient sample at 20:00 pm, 20120706 (c) ambient sample at 6:00 am, 20120709. The gray lines indicate points during the oxygen-free (He only) phase of the analysis, the blue line is for points during the oxidizing stage ($He+O_2$) of the analysis and the black line is for the points during the calibration stage ($CH_4+O_2$). The red line is a best-fit polynomial through the $CH_4+O_2$ points, while the dashed red lines are linear fits.





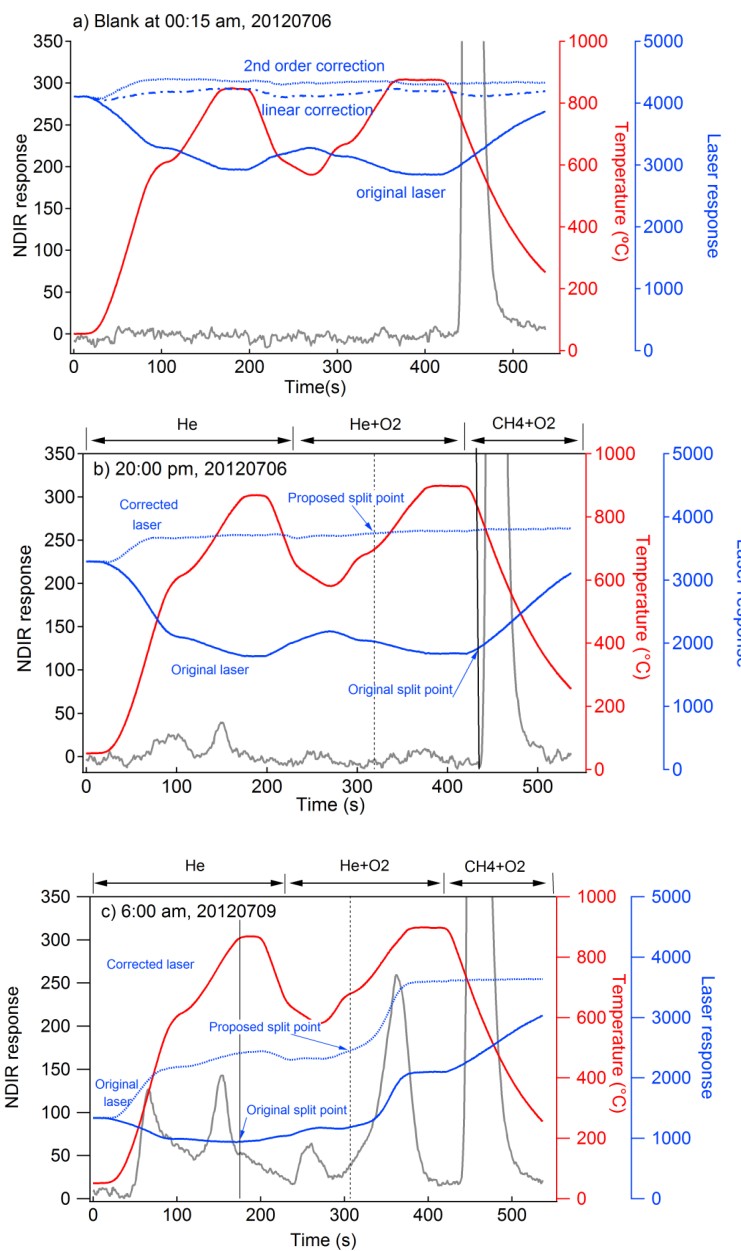

Figure A.2: Thermograms of selected Riyadh samples: (a) blank at 00:15 am, 20120706 (YYYYMMDD); (b) ambient sample at 20:00 pm, 20120706 with relatively low EC loading; (c) ambient sample at 6:00 am, 20120709 with relatively high EC loading.