# Peer review of "Sources of PM2.5 carbonaceous aerosol in Riyadh, Saudi Arabia"

_Atmospheric Chemistry and Physics, 2017_

## Referee Comment (RC1) · Anonymous Referee #1 · 12 Oct 2017

This manuscript describes comprehensive measurements of particulate organic and elemental carbon and metals in Riadyh, Saudia Arabia. There is detailed trajectory and emission/source analysis. There is investigation of the secondary component of OC with consideration of differing weekend/weekday emissions and the potential different chemical regimes (e.g., different NOx emissions with potential impact on secondary OC formation). This work is similar to studies and analyses performed for areas of the U.S. and Europe. This is dataset is novel for its location, and may provide useful constraints for atmospheric modeling studies applied to this region. My comments are mostly minor and should be considered prior to publication.

Abstract: If most of the OC is secondary and Ca is predominantly primary, how can they have similar sources (line 32)? Is it that the precursor VOC has a similar source

as Ca? Could it be that the species undergo similar atmospheric processing?

Starting at Line 70: Is atmospheric re-suspended dust due to wind or human activity? If it is due to human activity is it "anthropogenic"?

The chosen cities for comparison in Table 1 see randomly chosen, and this is probably not the case. Are these areas or their air quality similar to Riyadh in some way?

Starting at line 276 "Since OC concentrations had no significant weekday-weekend variation, the increase in OC/EC ratio during the weekend likely indicates the importance of regional photochemical sources of SOC, although decreased NOx emissions on weekends may promote more efficient photochemical processing of local SOC precursors" This is a curious conclusion to draw from Figures 5 and S2, but maybe not in the context of results discussed later in the manuscript. From these figures, it appears that changes in [EC] are what drive the changes in the OC/EC ratio predominantly. Perhaps the authors are intending to state primary OC emissions follow trends in EC and that the OC on Wednesdays is more primary than say on the weekends when SOC makes a larger contribution? It seems the authors allude to this when discussing figure 6. This needs to be shown first and the authors need to state the findings to support this statement more articulately.

Why not explore the diurnal profiles also separated by weekend/weekday. That would help support the statements the authors make (above) regarding OC/EC findings.

Is it possible that if Calcium carbonate forms, other compounds, for example would potassium carbonate form? It does seem from Figure 8 that there are two regimes for K vs. Ca. Does that inform the OC and Ca correlation analysis further?

Editorial: Sometimes the authors use present tense (e.g., line 80) and sometimes past tense (e.g., line 216, 218) and it is distracting.

---

## Referee Comment (RC2) · Anonymous Referee #2 · 5 Nov 2017

This is mostly an excellent submission, presenting results from a long-term measurement campaign in Saudi Arabia and source apportionment analysis. Questions I had were subsequently answered in the submission, which is usually a sign the authors have done a thorough job with the analysis.

However, one glaring flaw appears to be that in 2012, the weekend in Saudi Arabia was Thu-Fri, not Fri-Sat. The authors should re-analyze their data accordingly.

2013 news article about the weekend switch: http://english.ahram.org.eg/NewsContent/2/8/74730/World/Region/Saudi-Arabia-changes-working-week-to-SunThurs-Offi.aspx

Specific observations:

1.  How often or when was the OC/EC filter changed? Was the OC/EC correction

different at the beginning than at the end? Did the switch coincide with particular days of the week?

2. Lines 450-451 - the authors say the limited sample size means they can't quantify the local and regional contributions to OC and EC. However, this limitation only applies to the 24-hour metals analysis, which also appears to show that SOC, associated with Ca, may be regional. So couldn't the authors use the hourly-resolved OC/EC data to estimate local contributions to OC and EC?

3. Table 1 could be rearranged to list the current study next to the 2007 middle-east study, as that is most relevant to the present analysis. I would have liked to see a more extensive comparison of the two sets of results.

4. Figures 2 and 3 just have EC/OC concentrations as the axis title, but the OC/EC ratios are also shown. Maybe put the ratio on the secondary axis with an appropriate title? Also, OC/EC ratios in the 100s - admittedly outliers - are interesting. Are those associated with low pollution levels?

5. Figure 6(c) - Axis title is wrong. Also, the average ratios appear to be wrong, as almost all of them are higher than 75th percentile of the data. What do the caps represent - 90th or 95th or 99th percentile?

6. Figure 8 - the high correlation between OC and Ca appears driven by a single high-value sample. Is that really good enough to push the OC-Ca connection?

7. Figure 11 - what happened to the samples in August? Also, maybe the Aug 31 sample should be grouped with September?

8. Figure 11 - was there no cement or gas flare or local vehicular contributions in May? That seems inconceivable. The authors should explain a bit more.

9. Figure A.2 shows that all the corrected laser values increase in transmittance at the beginning, which is rather strange - no EC should have left the filter in He1! What is going on - is the correction wrong?

---

## Author Comment (AC1) · 17 Dec 2017

We thank the reviewers for their thoughtful reviews, which helped us to improve our manuscript. Point-by-point responses are provided as follows.

***Reviewer 1:***

Abstract: If most of the OC is secondary and Ca is predominantly primary, how can they have similar sources (line 32)? Is it that the precursor VOC has a similar source as Ca? Could it be that the species undergo similar atmospheric processing?

*Response:*
The reviewer's points are well taken. Accordingly, we have revised the sentence (line 29-31; note that the line number refers to the revised version of the manuscript) as follows. "Total OC was correlated with Ca ($R^2$ of 0.63), suggesting that OC precursors and Ca may have had similar sources, and the possibility that they underwent similar atmospheric processing."

Starting at Line 70: Is atmospheric re-suspended dust due to wind or human activity? If it is due to human activity is it "anthropogenic"?

*Response:*
It is likely that both mechanisms are important in controlling dust loadings. To remove the ambiguity, line 72 was revised as "In prior studies conducted in the Middle East, dust was identified as the major source of $PM_{10}$ (Givehchi, et al., 2013);"

The chosen cities for comparison in Table 1 seem randomly chosen, and this is probably not the case. Are these areas or their air quality similar to Riyadh in some way?

*Response:*
We sought observations of OC/EC measurements made in the last 10 years in urban areas world-wide. Urban areas may be hypothesized to have similar sources of PM, e.g. vehicular exhaust and industrial emissions, albeit that the nature of the PM will vary depending on factors such as local fleet and fuel mixes and local industry. Thus, the table title is revised as "Comparison of OC and EC concentrations ($\mu g\ m^{-3}$) measured in urban areas world-wide". In the revision, we added one more Korean urban city and replaced the US study with two large cities as follows below. We also compared our measurements with other studies in the Middle East expected to have similar climatological conditions as Riyadh.

Combining a response to this point and to Review 2, comment 3, the related main text (lines 237-254) was revised as follows:

"Table 1 presents some comparative values of measured EC and OC concentrations in $PM_{2.5}$ in urban areas world-wide, since urban areas are expected to share some anthropogenic source types (e.g. vehicular and industrial emissions) with Riyadh. The average concentrations in this work for both EC and OC were remarkably consistent with those reported by von Schneidemesser et al. (2010) and Abdeen et al. (2014) for 11

Middle Eastern sampling sites, including Tel Aviv, a major city in Israel (OC: 4.8 and EC: 1.6 µg m$^{-3}$). The average OC concentrations in Riyadh were also comparable to those reported for suburban Hong Kong (4.7 µg m$^{-3}$, Huang et al., 2014b), higher than Cleveland and Detroit, US (3.10 and 3.54 µg m$^{-3}$, Snyder et al., 2010), but lower than those reported for Gwangju, Korea (5.0 µg m$^{-3}$, Batmunkh et al., 2016),Veneto, Italy (5.5 µg m$^{-3}$, Khan et al., 2016), Athens, Greece (6.8 µg m$^{-3}$, Grivas et al., 2012), urban Hong Kong (10.1 µg m$^{-3}$, Ho et al., 2006), Delhi, Indian (16.5 ± 6.6 µg m$^{-3}$, Satsangi et al., 2012), and Beijing, China (18.2 ± 13.8 µg m$^{-3}$, Zhao et al., 2013), reflective of the different mix of sources and different photochemical environments. EC concentrations also vary widely among urban regions, depending on the characteristics of local sources."

| City | Duration | EC | OC | EC | OC | References |
|---|---|---|---|---|---|---|
| | | Conc. (µg m$^{-3}$) | | S.D. (µg m$^{-3}$) | | |
| Athens, Greece | Jan to Aug, 2003 | 2.2 | 6.8 | | | Grivas et al., 2012 |
| Gwangju, Korea | Winter of 2011 | 1.7 | 5.0 | 0.9 | 2.5 | Batmunkh et al., 2016 |
| Cleveland, US | Jul, 2007 and Jan, 2008 | 0.33 | 3.10 | 0.01 | 0.78 | Snyder et al., 2010 |
| Detroit, US | | 0.35 | 3.54 | 0.01 | 0.86 | |
| Beijing, China | Selective days in four seasons from 2009 to 2010 | 6.3 | 18.2 | 2.9 | 13.8 | Zhao et al., 2013 |
| Urban, Hong Kong | Nov, 2000 to Feb, 2001 and Jun, 2001 to Aug, 2001 | 5.71 | 10.12 | 0.89 | 1.92 | Ho et al., 2006 |
| Suburban, Hong Kong | Mar, 2011 to Feb, 2012 | 0.86 | 4.7 | 0.53 | 2.87 | Huang et al., 2014b |
| Veneto, Italy | Apr 2012 to Feb 2013 | 1.3 | 5.5 | | | Khan et al., 2016 |
| Delhi, India | Dec 20, 2012 to Feb 26, 2013 | 12.04 | 16.46 | 4.43 | 6.61 | Panda et al., 2016 |
| Middle East | Jan to Dec, 2007 | 2.1 | 5.3 | 2.2 | 4 | von Schneidemesser, et al., 2010 Abdeen, et al., 2014 |
| (11 sampling sites in Palestine, Jordan and Israel) | | | | | | |
| Riyadh, Saudi Arabia | Apr to Sep, 2012 | 2.13 | 4.76 | 2.52 | 4.4 | this study |

References added:
Batmunkh, T., Lee, K., Kim, Y. J., Bae, M.-S., Maskey, S., Park, K.: Optical and thermal characteristics of carbonaceous aerosols measured at an urban site in Gwangju, Korea, in the winter of 2011, J. Air & Waste Manage Association, 66, 151-163, DOI: 10.1080/10962247.2015.1101031, 2016.
Snyder, D. C., Rutter, A. P., Worley, C., Olson, M., Plourde, A., Bader, R. C., Dallmann, T., Schauer, J. J.: Spatial variability of carbonaceous aerosols and associated source tracers in two cities in the Midwestern United States, Atmos. Environ., 44, 1597-1608, 2010.

von Schneidemesser, E., Zhou, J., Stone, E. A., Schauer, J. J., Qasrawi, R., Abdeen, Z., Shpund, J., Vanger, A., Sharf, G., Moise, T., Brenner, S., Nassar, K., Saleh, R., Al-Mahasneh., Q. M., Sarnat, J.A.: Seasonal and spatial trends in the sources of fine particle organic carbon in Israel, Jordan, and Palestine, Atmos. Environ., 44, 3669-3678, 2010.

Starting at line 276 "Since OC concentrations had no significant weekday-weekend variation, the increase in OC/EC ratio during the weekend likely indicates the importance of regional photochemical sources of SOC, although decreased NOx emissions on weekends may promote more efficient photochemical processing of local SOC precursors" This is a curious conclusion to draw from Figures 5 and S2, but maybe not in the context of results discussed later in the manuscript. From these figures, it appears that changes in [EC] are what drive the changes in the OC/EC ratio predominantly. Perhaps the authors are intending to state primary OC emissions follow trends in EC and that the OC on Wednesdays is more primary than say on the weekends when SOC makes a larger contribution? It seems the authors allude to this when discussing figure 6. This needs to be shown first and the authors need to state the findings to support this statement more articulately.

*Response:*
Thank you for these helpful comments. We switched the order of sections 3.2 and 3.3 and "Diurnal variation of OC and EC" now comes before "Weekend effect in OC and EC concentrations", as suggested.

Also, regarding this sentence:
"Since OC concentrations had no significant weekday-weekend variation, the increase in OC/EC ratio during the weekend likely indicates the importance of regional photochemical sources of SOC, although decreased $NO_x$ emissions on weekends may promote more efficient photochemical processing of local SOC precursors (Gentner et al., 2012)." we have modified it as follows:
"OC concentrations had no significant weekday-weekend variation. The decrease of EC was the main driver of the increasing OC/EC ratio during the weekends, indicating reduced primary emissions and effective SOC formation / transport during the weekends."

Why not explore the diurnal profiles also separated by weekend/weekday. That would help support the statements the authors make (above) regarding OC/EC findings.

*Response:*
We replaced the original Fig. S3 with the following figure and added the corresponding discussion in lines 276 to 277:
"The diurnal variations of OC and EC on weekdays and weekends exhibited similar trends (Fig. S3), but EC was higher during weekdays."
Lines 345 to350:
"The diurnal variations of POC and SOC were similar on weekdays and weekends, but the weekday-to-weekend changes in POC and SOC had opposite trends. The estimated POC was 2.2±2.5 µg m$^{-3}$ on weekdays and decreased to 1.5±1.9 µg m$^{-3}$ on weekends. The estimated SOC was 2.6±2.9 µg m$^{-3}$ on weekdays and increased by 23% to 3.2±4.5

µg m$^{-3}$ on weekends. The elevated SOC during weekends was likely due to regional production and transport."

[Figure]

Is it possible that if calcium carbonate forms, other compounds, for example would potassium carbonate form? It does seem from Figure 8 that there are two regimes for K vs. Ca. Does that inform the OC and Ca correlation analysis further?

*Response:*
Yes, it is possible that other carbonates were present. We revised lines 377 to 380 as follows:
"The correlation between Ca and other dust metal species (Al, Fe, K, Fe and Mg), however, showed two divergent regimes, suggestive of an additional Ca-containing source besides dust, that may have shared the same sources as OC."

Editorial: Sometimes the authors use present tense (e.g., line 80) and sometimes past tense (e.g., line 216, 218) and it is distracting.

*Response:*
Revised accordingly.

---

## Author Comment (AC2) · 17 Dec 2017

We thank the reviewers for their thoughtful reviews, which helped us to improve our manuscript. Point-by-point responses are provided as follows.

**Reviewer 2:**

This is mostly an excellent submission, presenting results from a long-term measurement campaign in Saudi Arabia and source apportionment analysis. Questions I had were subsequently answered in the submission, which is usually a sign the authors have done a thorough job with the analysis. However, one glaring flaw appears to be that in 2012, the weekend in Saudi Arabia was Thu-Fri, not Fri-Sat. The authors should re-analyze their data accordingly. 2013 news article about the weekend switch: http://english.ahram.org.eg/NewsContent/2/8/74730/World/Region/Saudi-Arabia-changes-working-week-to-SunThurs-Offi.aspx

*Response:*
Thank you to the reviewer for pointing out this flaw.
Upon re-analyzing the data for the correct weekend dates, we discovered that the analysis was correct (i.e., the weekend was defined as Thu-Fri) but the labels in the figure were incorrect. The axis label in Fig. 5 (now Fig. 6) was updated as shown below. The main text (lines 289-293) has also been corrected:
"To investigate whether a weekend effect could be discerned in the Riyadh dataset, two-sample t-tests assuming unequal variances were performed for hourly EC and OC samples, grouped according to whether they were obtained on weekdays (Saturday to Wednesday) or on weekends (Thursday and Friday)."

[Figure]

1. How often or when was the OC/EC filter changed? Was the OC/EC correction different at the beginning than at the end? Did the switch coincide with particular days of the week?

*Response:*
OC/EC filters were changed after the laser intensity was reduced to 2000 or 3000 [a.u.] We have added a footnote to Table S1 to indicate this. The time to reach this threshold was mainly controlled by particulate concentrations, especially during dust storms. Some

filters lasted for only 24-48 hours, while some lasted for about 6-7 days. The OC/EC correction was the same over the entire filter lifetime, as long as there were refractive particles on the filters.

2. Lines 450-451 - the authors say the limited sample size means they can't quantify the local and regional contributions to OC and EC. However, this limitation only applies to the 24-hour metals analysis, which also appears to show that SOC, associated with Ca, may be regional. So couldn't the authors use the hourly-resolved OC/EC data to estimate local contributions to OC and EC?

*Response:*
We removed this sentence.

3. Table 1 could be rearranged to list the current study next to the 2007 middle-east study, as that is most relevant to the present analysis. I would have liked to see a more extensive comparison of the two sets of results.

*Response:*
We have rearranged the table as suggested, revised the discussion of Table 1, lines 275-289.

4. Figures 2 and 3 just have EC/OC concentrations as the axis title, but the OC/EC ratios are also shown. Maybe put the ratio on the secondary axis with an appropriate title? Also, OC/EC ratios in the 100s - admittedly outliers - are interesting. Are those associated with low pollution levels?

*Response:*
Figs. 2 and 3 have been modified as suggested.
Most of the high OC/EC ratios (>100) were caused by rapid, large increases in OC and an accompanying decrease in EC during the measurement. An example is shown below, covering two consecutive days of data. On 2012/04/19, OC was $6.24 \pm 2.27$ µg m$^{-3}$, EC was $2.08 \pm 1.79$ µg m$^{-3}$, and OC/EC ratio was $4.11 \pm 2.07$; on 2012/04/20, OC increased to $16.09 \pm 12.20$ µg m$^{-3}$, EC decreased to $0.71 \pm 0.51$ µg m$^{-3}$, and the OC/EC ratio increased to $39.80 \pm 37.37$. The highest OC/EC ratio on 2012/04/20 was 102. The total carbon on the second day was almost double that of the first day. This event may be a dust plume, as the methodology applied in the study cannot correct for carbonate interference in the OC/EC observations, although we have made inferences as to the presence of carbonate, as described in the text. While we do not have additional data to fully explore the causes of these excursions, we have no firm reason to remove these data, and thus we kept them in our dataset.

[Figure]

5. Figure 6(c) - Axis title is wrong. Also, the average ratios appear to be wrong, as almost all of them are higher than 75th percentile of the data. What do the caps represent - 90th or 95th or 99th percentile?

*Response:*
Thank you for pointing this out; we have corrected the y-axis title. The upper and lower caps represent 90th and 10th percentiles, respectively. The high average OC/EC ratios were caused by several individual high OC/EC ratios, as explained in comment 4. We retained the median and removed the point indicating the average in Fig. 5 (now Fig 6) to avoid confusion, and made similar changes to the other box-and-whisker figures.

6. Figure 8 - the high correlation between OC and Ca appears driven by a single high-value sample. Is that really good enough to push the OC-Ca connection?

*Response:*
Removing the high-value data point indeed decreased $R^2$, however, the time series of OC and Ca matched each other well and better than that of EC and Ca (see the figures below). The relationship between OC and Ca looks to be real, but what caused the relationship was uncertain as we discussed in the main text: whether a methodology artifact or that these species actually shared the same source origins. We have included these time series in the supporting material to illustrate this relationship and modified the sentence 368-370 as follows "However, OC had a relatively strong correlation with Ca ($R^2$ of 0.63) (Fig. 8 and Fig. S7) but, similar to EC, a poor correlation with other dust species (not shown)."

[Figure]

7. Figure 11 - what happened to the samples in August? Also, maybe the Aug 31 sample should be grouped with September?

*Response:*

In 2012, Ramadan and Eid ran from July 20-August 24 and no measurements were made during this period. Thank you for the suggestion to regroup the Aug 31 sample with the September samples. We have done this and updated the numbers in lines 474 to 478. "The contribution of the mixed source ranged from 37% in May (0.7 µg m$^{-3}$) to 95% in September (6.7 µg m$^{-3}$). The EC concentration was also mainly attributed to the mixed source (1.9 µg m$^{-3}$, 92%)."

8. Figure 11 - was there no cement or gas flare or local vehicular contributions in May? That seems inconceivable. The authors should explain a bit more.

*Response:*

EC was indicated as a tracer for the mixed sources and Pb was the tracer for long-range transport, according to the PMF-resolved source profile (Figure 10). Daily-average EC concentrations were relatively low while Pb concentrations were high in May samples compared with those in the other periods, as shown in the following figure. This suggests that long range transport was dominant at that time. Due to the limited number of samples, PMF was unable to pick up the low contribution from mixed sources, with the result that minimal cement / gas flare / local vehicular contributions were found for most May samples. Lines 478 to 480 are revised as follows, "In some May samples, the mixed source contribution was negligible, as the source tracer, EC, was only 0.1-0.4 µg m$^{-3}$, about one order of magnitude lower than that in other periods. The tracer analysis suggested that long-range transport was dominant for those samples."

[Figure]

9. Figure A.2 shows that all the corrected laser values increase in transmittance at the beginning, which is rather strange - no EC should have left the filter in He1! What is going on - is the correction wrong?

*Response:*

Figure A.2a shows a thermogram for one of the "aged" blank samples. As expected, negligible carbon was detected in this sample, but the refractive particles that remained on the filter from previous samples still significantly influenced the laser variation. Fig. A.1a shows that the quadratic fit could not reproduce the signal at low temperature (<200 °C), so the increasing corrected laser signal during the He1 and He2 phases is likely an artifact of the correction methodology, and may not indicate EC evolution. However, we note that the studies of Wang et al. (2010) and Bladt et al. (2012) showed that refractory metal oxides / salts may cause premature EC evolution in an inert environment, thereby increasing the laser signal. A high loading of dust metals on the filters in our Riyadh samples may thus have caused premature EC evolution during the He stage. If this were the case, the original variable laser signal that we showed was

dependent on temperature may have masked that phenomenon, while the corrected one revealed it.

We revised the main text from lines 561 to 570as follows:

"It is noted that although the quadratic equation correction produced a better laser signal for purposes of the carbon analyses, this correction did not work perfectly in the low temperature He phase, where the corrected laser signals exhibited unexpected increases. However, this shortcoming did not substantially influence the accuracy of the correction during subsequent carbon evolution. We note that premature evolution of EC, leading to an increasing laser signal in the inert environment due to the existence of refractory metal oxides, was observed in the studies of Wang et al. (2010) and Bladt et al. (2012). The increases in the corrected laser signal during the He stage in this study may be partially due to the same cause, as Riyadh samples contained abundant metal oxides."

Reference:

Bladt, H., Schmid, J., Kireeva, E.D., Popovicheva, O. B., Perseantseva, N. M., Timofeev, M. A., Heister, K., Uihlein, J., Ivleva, N. P., Niessner, R: Impact of Fe Content in Laboratory-Produced Soot Aerosol on its Composition, Structure, and Thermo-Chemical Properties, Aerosol Sci. Tech., 46, 1337-1348, DOI:10. 1080/02786826.2012.711917, 2012.

Wang, Y., Chung A., Paulson, S.E.: The effect of metal salts on quantification of elemental and organic carbon in diesel exhaust particles using thermal-optical evolved gas analysis, Atmos. Chem. and Phys., 10, 11447-11457, 2010.

---

## Author Response (AR2)

We thank the reviewer for the suggestions to improve the manuscript. Our responses to the two comments are provided as follows.

Two minor revisions are suggested:
1) The authors say that they are unable to separate local vehicular emissions from the mixed source, but they also find that EC emissions are lower during the weekend. I would argue that is a good indicator of local traffic emissions, even if not all of it (some people still drive on the weekends.) Gas flares and cement industries probably are active all week long, thus the weekday/weekend difference should indicate local sources.

Response:

We modified line 290 to 295 as follows: "This reduction may be ascribed to the decrease in the local vehicular activities and industrial activities during the weekend. Therefore, local EC can be roughly estimated to be 0.51 $\mu g\ m^{-3}$, about 22% of total EC, by the difference in average EC concentrations between weekday and weekend. As there were still some local traffic and industrial activities during the weekend, this estimate is likely a lower bound of the local contribution to EC concentrations in the study. " We also added one sentence near line 474 of the conclusion, "at least 22% of EC was ascribed to local sources".

2) The authors should check if the low EC in May samples is due to premature EC evolution due to increased refractory matter on the filter substrate. If not, they could add a statement to that effect.

Response:

We checked the thermograms of May samples and most of the samples are similar to Fig. A. 2 (b), shown in the following figure. Unlike Fig. A. 2 (c), where the incorrect split point can be seen clearly in He phase, it is not possible to state that premature EC evolution did or did not occur. In general, however, the sum of OC+EC was low in May.

[revised manuscript text omitted]